# The Biological and Clinical Role of the Telomerase Reverse Transcriptase Gene in Glioblastoma: A Potential Therapeutic Target?

**DOI:** 10.3390/cells13010044

**Published:** 2023-12-25

**Authors:** Vincenzo Di Nunno, Marta Aprile, Stefania Bartolini, Lidia Gatto, Alicia Tosoni, Lucia Ranieri, Dario De Biase, Sofia Asioli, Enrico Franceschi

**Affiliations:** 1Nervous System Medical Oncology Department, IRCCS Istituto delle Scienze Neurologiche di Bologna, 40139 Bologna, Italyenricofra@yahoo.it (E.F.); 2Department of Experimental, Diagnostic and Specialty Medicine, University of Bologna, 40138 Bologna, Italy; 3Department of Oncology, Azienda Unità Sanitaria Locale (AUSL) Bologna, 40139 Bologna, Italy; 4Solid Tumor Molecular Pathology Laboratory, IRCCS Azienda Ospedaliero-Universitaria di Bologna, 40138 Bologna, Italy; 5Department of Pharmacy and Biotechnology (FaBit), University of Bologna, 40126 Bologna, Italy; 6IRCCS Istituto delle Scienze Neurologiche di Bologna, 40139 Bologna, Italy; 7Department of Biomedical and Neuromotor Sciences (DIBINEM), Surgical Pathology Section, Alma Mater Studiorum, University of Bologna, 40126 Bologna, Italy

**Keywords:** TERT, TERT inhibitor, glioma, glioblastoma

## Abstract

Glioblastoma *IDH*-wildtype represents the most lethal and frequent primary tumor of the central nervous system. Thanks to important scientific efforts, we can now investigate its deep genomic assessment, elucidating mutated genes and altered biological mechanisms in addition to its clinical aggressiveness. The telomerase reverse transcriptase gene (*TERT*) is the most frequently altered gene in solid tumors, including brain tumors and GBM *IDH*-wildtype. In particular, it can be observed in approximately 80–90% of GBM *IDH*-wildtype cases. Its clonal distribution on almost all cancer cells makes this gene an optimal target. However, the research of effective TERT inhibitors is complicated by several biological and clinical obstacles which can be only partially surmounted. Very recently, novel immunological approaches leading to *TERT* inhibition have been investigated, offering the potential to develop an effective target for this altered protein. Here, we perform a narrative review investigating the biological role of TERT alterations on glioblastoma and the principal obstacles associated with TERT inhibitions in this population. Moreover, we discuss possible combination treatment strategies to overcome these limitations.

## 1. Introduction

Telomeres are repeated nucleotide sequences located at the chromosomal extremities. Telomere erosion due to multiple cell divisions finally triggers a DNA damage response, replicative senescence, and growth arrest in somatic cells. Not surprisingly, the preservation of telomere integrity is a critical hallmark that cancer cells must acquire to become “immortal”. Tumor cells have two different methods to maintain telomere integrity. These are represented by telomerase reverse transcriptase (*TERT*) alterations and a telomerase-independent mechanism called “alternative lengthening of telomeres” (ALT) that depends on *ATRX/DAXX* (a-thalassemia/mental retardation syndrome X-linked/Death-associated protein 6) complex. The “alternative lengthening of telomeres” (ALT), which depends on the *ATRX/DAXX* complex, is mutually exclusive with TERT in gliomas [1]. This last mechanism is a type of homologous recombination called break-induced telomere synthesis. Normally, homologous recombination is necessary to repair broken DNA strands by adding nucleotides that are complementary to the undamaged DNA segment. This same mechanism is used to extend telomeres [1].

*TERT* promoter mutations are the most frequent non-coding hotspot alteration in human cancers [2]. They have been described in several tumor entities, comprising hepatocellular carcinomas [3], urothelial carcinomas of the bladder [4], and melanomas, where they were described for the first time [5]. 

The fact that *TERT* alterations are shared by several different malignancies reflects the importance of this gene for cancer. Physiologically, *TERT* encodes for the catalytic subunit of a ribonucleoprotein complex responsible for telomere maintenance called “telomerase”. Conversely, *TERT* promoter (*TERTp*) variants or *TERT* overexpression are associated with enhanced telomerase activity and cell immortalization [6].

TERT promoter mutations are a very frequent event in central nervous system tumors (CNS) [7], especially in gliomas [8,9,10]. In particular, in patients with GBM *IDH*-wildtype, TERT alterations could reach 80–90% incidence [8]. Furthermore, TERT promoter mutations are an early genetic event in gliomagenesis, explaining the observed homogeneous distribution among cellular tumor subclones [11]. *IDH*-wildtype glioblastoma, according to the Central Nervous System World Health Organization Classification (CNS WHO), 5th edition [12], is a diffuse, astrocytic glioma with wild-type *IDH* and H3, exhibiting one or more of the following histological or genetic features: microvascular proliferation, necrosis, TERT promoter mutation, EGFR gene amplification, and +7/−10 chromosome copy-number changes (CNS WHO grade 4).

Considering the prevalence and incidence of this genomic alteration, the possibility of developing specific inhibitors for *TERT*-altered tumors is a very attractive opportunity for clinical oncology. Thanks to its ubiquitous distribution and elevated incidence, *TERT* appears as the perfect potential target in glioblastomas. However, several factors are limiting *TERT* inhibition, and, to date, no experimental compounds targeting *TERT* have been approved. Several strategies to inhibit *TERT* have been tested across multiple cancer types, comprising small molecule inhibitors and *TERT*-based immunotherapy, in particular, vaccines.

This review aims to provide an update of the evidence on the biology and physiopathology of *TERT*, exploring promising novel strategies and limitations to TERT inhibition in patients with glioblastomas.

## 2. TERT Role and Implications in Tumors

Telomerase is expressed in stem cells of proliferative tissues such as blood and skin [13]. When activating hot-spot mutations in the promoter region of *TERT (pTERT)* occur, these result in an upregulation of telomerase complex activity and thus constitute a relevant mechanism for the immortalization of tumor cells. This makes *pTERT* mutations one of the most common alterations shared in solid malignancies, including CNS primary tumors.

In tumors, re-activation of telomerase and telomere length maintenance constitute a key step in tumorigenesis. As previously reported, telomeres are guanine (G)-rich nucleotide repeats at the end of chromosomes. Without specific maintenance mechanisms, telomeres progressively shorten with mitotic cell divisions due to the intrinsic properties of DNA replication machinery. This is known as an “end replication problem”. The sequence loss at the end of chromosomes is consecutively responsible for cellular senescence or a process of cell death known as a “telomere crisis”.

Telomerase is recruited to the 3′ tail of telomeric DNA by the shelterin complex, which consists of six proteins that are involved in the activation of the whole enzymatic complex: telomeric repeat-binding factor 1 (TRF1), TRF2, Ras-related protein 1 (RAP1), TRF1-interacting nuclear factor 2 (TIN2), tripeptidyl peptidase 1 (TPP1), and protection of telomeres protein 1 (POT1) (Table 1 and Figure 1) [14]. Among the shelterins, TPP1 has a key role in recruiting telomerase to telomeres and activating telomere synthesis.

Indeed, TPP1 engages POT1. The POT1 binds the single-stranded telomeric DNA, and TIN2. The TIN2 interacts with the double-stranded telomeric DNA through TRF1 and TRF2 [14].

The role of TERT is to directly prevent this mechanism through the elongation of telomeres. In particular, TERT acts as a reverse transcriptase and employs its internal RNA molecule (hTR) as a template to add hexameric 5′-TTAGGG-3′ tandem repeats at chromosomal ends. Telomeric DNA consists of both a double-stranded DNA and a single-stranded DNA.

TPP1 engages POT1, which binds the single-stranded telomeric DNA, and TIN2. This last interacts with the double-stranded telomeric DNA through TRF1 and TRF2 [14]. Recently, a better definition of the cryo-electron microscopy structures of TERT and hTR allowed a clearer interpretation of the TERT–hTR interaction, and the TERT–TPP1 interaction, providing new potential drug targets [15].

**Table 1 cells-13-00044-t001:** Principal alterations leading to telomerase reverse transcriptase (*TERT*) gain of function and a summary of TERT principal biological activities. ETS = E26 transformation-specific family transcription factors; POT1 = protection of telomeres protein 1; RAP1 = Ras-related protein 1; TIN2 = TRF1-interacting nuclear factor 2; TPP1 = Tripeptidyl peptidase 1; TRF 1 = telomeric repeat-binding factor 1; TRF2 = telomeric repeat-binding factor 2 [14].

*TERT* Alterations	*TERT* Function
Hotspot mutations of the *TERT* promoter gene result in *TERT* hyperexpression or hyperactivation [14].	Telomere elongation by telomerase activity. This function is mediated by six different proteins: (TRF1), TRF2, RAP1, TIN2, TPP1, and POT1 [14]. All these proteins constitute the shelterins complex.
Increased *TERT* m-RNA expression mediated by *TERT* promoter mutations and ETS interaction mediated by the GA-binding protein (GABP).	Repression of growth inhibitory factors [16].
	Resistance to apoptosis diminished the capacity for DNA repair [17,18].

Besides its central role in telomere length maintenance, we have increasing data about telomere length-independent functions of *TERT*. In particular, *TERT* enhances cell proliferation through repression of growth inhibitory factors [16], impairment of DNA damage responses, and resistance to apoptosis [17,18].

Overall, *TERT* expression is regulated by multiple factors on genetic and epigenetic levels, including promoter mutations, promoter methylations, chromosomal rearrangements, and amplifications [19].

On a transcriptional level, a *TERTp* mutation implies the creation of transcriptionally active mutant promoters, such as a novel binding site for E26 transformation-specific (ETS) family transcription factors. The E26 transcription factors facilitate TERT mRNA expression. The GA-binding protein (GABP) has been identified as the only ETS transcription factor able to bind the mutated ETS motif [20].

The GA-binding protein transcription factor subunit alpha (GABPα) is a multimer made of two kinds of subunits that can selectively bind to the mutant *TERT* promoter: GABPα (a DNA-binding subunit), and GABPβ (a transactivating subunit). GABPβ exists in two alternative paralogues, GABPβ1 and GABPβ2. GABPβ1L (GABPβ1 long), a potential druggable target, is one of the two isoforms of GABPβ1. The isoform GABPβ1S (GABPβ1 short) differs from GABPβ1L due to the different site of GABPα binding.

The β1S can dimerize with GABPα, and both β1L and β2 have a leucine-zipper domain that mediates the tetramerization of two GABPαβ heterodimers. Among ETS transcription factors, GABP is the only one able to bind neighboring native ETS motifs and mutant ETS motifs as a heterotetrametric complex [21]. The GABP tetramer-forming isoforms are critical in activating the mutant TERT promoter; for this reason, these are under evaluation as potential therapeutic targets (Figure 2).

## 3. *TERTp* Mutations in GBM *IDH*-Wildtype

In the last two decades, genome-wide sequencing technologies have allowed a deep molecular characterization of neoplasms. Glioblastoma (both *IDH*-wildtype and, now, reclassifying astrocytoma *IDH*-mutant grade 4 CNS WHO) was the first cancer studied within the Cancer Genome Atlas Program (TCGA), whose aim was to list and describe the major cancer-causing genome alterations [22]. Novel evidence reported *TERTp* mutations as the most frequent genetic event in GBM *IDH*-wildtype.

*TERTp* mutations are mainly transitions caused by the substitution of the pyrimidine nucleotide cytidine with thymidine (C>T). The C228T (c.–124C>T, g.1295228 on GRCh37) and C250T (c.–146C>T, g.1295250 on GRCh37) mutations occur upstream of the transcriptional start site and represent the most frequent hotspot *TERTp* mutations. Both transitions generate an identical 11bp sequence which constitutes a novel ETS binding motif.

A different mutational position in *TERT* promoters is linked to a different *TERT* mutated expression. Indeed, C228T and C250T mutations show a 14-fold and 7-fold increase in mRNA expression, respectively [5]. 

Apart from GBM *IDH*-wildtype, *TERTp* mutations have been described in almost 100% of oligodendrogliomas, 80–90% of molecular/non-molecular GBM *IDH*-wildtype, and 7% of *IDH* mutant astrocytomas [10,12,23]. Overall, the C228T mutation has been reported with a higher frequency than the C250T mutation.

Among *TERTp*-mutated gliomas, a strong association has been observed with older age [1,24]. *TERTp* mutations are more frequent in adults than in pediatric patients [25]. *TERTp* mutation and ALT (secondary to *ATRX* mutation) are complementary mechanisms for telomere lengthening in GBM *IDH*-wildtype and are mutually exclusive.

Several studies investigated the prognostic role for *TERTp* in GBM *IDH*-wildtype patients with conflicting results (Table 2) [23,26,27,28,29,30,31,32,33,34]. After the advent of the Central Nervous System World Health Organization Classification (CNS WHO), 5th edition, *TERTp* should be considered an essential factor for the molecular diagnosis of GBM *IDH*-wildtype without a validated prognostic role [35]. Moreover, it has been observed as a co-occurrence with the chromosome 7 gain and chromosome 10 loss (+7/−10) [1].

Chromosomal abnormalities involving genes that drive proliferation, such as *EGFR* and *PDGFA* on chromosome 7, probably happen earlier than *TERT*p mutations in GBM *IDH*-wildtype. Subsequently, *TERT*p mutations could be necessary later for the clonal expansion of cancer cells [36].

Further studies have focused on the relationship between *TERT* regulation and oncogenic pathways involved in cell proliferation. The B Raf proto-oncogene p.Val600Glu mutation (*BRAF* p.V600E) can induce *TERT* upregulation in tumors; however, this genetic event is extremely uncommon in GBM *IDH*-wildtype [28]. Focusing on the more frequent *EGFR* activation, reported in approximately 57% of GBM *IDH*-wildtype, McKinney et al. recently demonstrated that GABP receives signals from EGFR through AMP-activated protein kinase (AMPK). On the other hand, GABP binds *TERT* and activates telomerase. The authors hypothesized targeting EGFR may therefore decrease the activity of mutant *TERTp*, in particular, combination therapies targeting EGFR could downregulate proliferation and reduce *TERT* activity [37].

Finally, in diffuse astrocytic glioma adult type, *IDH*-wildtype, and H3-wildtype, the *TERTp* mutation seemed to be associated with the poorest prognosis. Ceccarelli et al. analyzed TCGA diffuse gliomas and found mutations of *TERTp* in 85% of cases. They confirmed significant *TERT* upregulation in *TERTp* mutant cases [1].

Berzero et al. [38] demonstrated that patients with strictly defined astrocytoma *IDH*-wt grade 2 with isolated pTERTmut do not have the same prognosis as those with glioblastoma *IDH*-wt. Giannini C and Giangaspero F highlighted that clinicians and pathologists should be aware of these conclusions [39]. A *TERT* mutation identification could not be sufficient to assume that the tumor will behave as glioblastoma, *IDH*-wildtype (WHO CNS grade 4) as proposed in the cIMPACT-NOW update 6 [40] and it may be too late for the results of this paper to be incorporated in the upcoming 2021 WHO classification for CNS Tumor. Indeed, a novel type of *IDH*-wildtype glioma is characterized by gliomatosis cerebri-like growth pattern, *TERT* promoter mutation, and distinct epigenetic profile, as recently described by Meuch A. et al. [41]. The patients’ outcome in this study was better compared to *IDH*-wildtype glioblastomas, with a median progression-free survival of 58 months and overall survival of 74 months (both *p*-value < 0.0001). Therefore, the identification of pTERT mutation is an important step in the diagnostic and prognostic predictive process. However, its effective role in this setting should be pondered considering other molecular alterations identified as well as other histopathological, clinical, and neuroradiological features of the disease.

**Table 2 cells-13-00044-t002:** Studies assessing the prognostic role of telomerase reverse transcriptase (*TERT*) gene mutations in GBM *IDH*-wildtype. To date, *TERT* should not be considered a prognostic factor in glioblastoma.

Study Connotation	Population of Study (*n*)	Frequency of *TERTp* Mutation	Impact of *TERTp* Mutations on Survival Outcome in GBM *IDH*-Wildtype	Additional Results
Nonoguchi (2013) [33]	358 GBM (*n* = 322 primary GBM *IDH*-wildtype; *n* = 36 secondary GBM)	55% (58% in primary GBM *IDH*-wildtype and 28% in secondary GBM)	Shorter survival in both univariate and multivariate analysis after adjustment for age and gender. However, no difference in survival in multivariate analyses after adjusting for other genetic alterations, or when primary and secondary GBM were separately analyzed	Positive correlation between *TERTp* and *EGFR* amplification, inverse correlations with *IDH1* mutations and *TP53* mutations.
Labussière M (2014) [27]	395 GBM *IDH*-wildtype	76%	Shorter PFS and OS	The absence of both *TERT*p mutation and *EGFR* amplification is associated with longer survival in patients with GBM.
Mosrati M (2015) [29]	92 GBM *IDH*-wildtype	86%	Shorter OS	*TERT* SNPs rs2736100 and rs10069690 correlate with an increased risk of GBM.
Spiegl-Kreinecker S (2015) [30]	126 GBM (*n*= 120 GBM *IDH*-wildtype; *n* = 6 IDH1 mutated)	73%	Shorter OS	*TERT* SNP rs2853669 improves survival in wt*TERTp* GBM *IDH*-wildtype. The shortest OS was detected in *TERTp-mutated* GBM *IDH*-wildtype with homozygous rs2853669 alleles.
Simon M (2015) [24]	192 GBM (*n* = 178 primary GBMGBM *IDH*-wildtype; *n* = 14 secondary GBM)	77% (80% in primary GBM; 28% in secondary GBM)	Shorter OS in all primary GBM	Poorer survival in patients with primary GBM *IDH*-wildtype and *TERTp* mutations who did not carry the variant G-allele for the rs2853669 polymorphism
Nguyen NH (2017) [32]	303	75%	No impact on OS	MGMT methylated patients showed improved survival only in the presence of *TERTp* mutation (analogous result in the cohort from TCGA).
Shu C (2018) [34]	304 GBM (273 GBM *IDH*-wildtype)	66%	No impact on OS	The subgroup with both unmethylated MGMT promoter and *TERTp* mutation had the worst prognosis. The main factors affecting survival in this group were age and Ki-67 positivity.
Brito C (2019) [31]	256 GBM (*n* = 245 GBM *IDH*-wildtype; *n* = 11 *IDH* mut.)	88% in GBM *IDH*-wildtype; 25% in IDH mut.	No impact on OS in GBM *IDH*-wildtype	*PTEN* favorable prognostic factor in GBM *IDH*-wildtype and unfavorable for astrocytoma *IDH*-wildtype.
Kikuchi Z (2020) [26]	147 GBM *IDH*-wildtype	62%	Shorter PFS and OS	*TERT*p mutant GBM *IDH*-wildtype is associated with multifocal/distant lesions.
Berzero G (2021) [38]	47 diffuse astrocytomas *IDH* wildtype	51%	Patients meeting criteria for molecular GBM had a shorter OS compared to those with gliomas not meeting molecular GBM criteria (42 vs. 57 months)	Patients with isolated TERT promoter mutation (16/26) had a more favorable outcome (median OS 88 months).
Muench A. (2023) [41]	16 patients with diffuse glioma *IDH*-wildtype TERT mutated.	TERT mutation in 12/15 cases.	Patients gliomatosis cerebri-like growth pattern.	Median progression-free survival of 58 months and overall survival of 74 months.

GBM *IDH*-wildtype without *TERTp* mutations (10–20%) showed *ATRX*-mutation and SWI/SNF-related matrix-associated actin-dependent regulator of chromatin subfamily A-like protein 1 (*SMARCAL1*) as mechanisms of ALT [19].

Some studies have investigated the potential predictive response of *TERT* mutations. In a metanalysis by Vuong HG et al., *MGMT* promoter methylation was related to a survival benefit in patients with *TERT*-mutated GBM but not in *TERT*-wt GBM *IDH*-wildtype receiving temozolomide [42]. However, it should be explained that this analysis also involved studies enrolling patients with *IDH*-mutated tumors previously defined as secondary GBM. Thus, the presence of a TERT mutation should not be considered a prognostic factor. Finally, some studies have focused on the investigation of specific *TERT* single nucleotide polymorphisms (SNPs). These studies have suggested that some SNPs could be associated with increased clinical aggressiveness (rs2736100) [29]. On the other hand, the polymorphism rs2853669 has been largely detected in lower-grade gliomas and is associated with a decreased *TERT* expression since it leads to the disruption of the *ETS2* binding sequence [27,43,44]. Again, the majority of these restricted the analysis on patients with WHO-CNS 5th-defined *IDH*-wildtype glioblastoma. Giunco S et al. confirmed the possible predictive role of the re2853669 in a cohort of patients with *IDH*-wildtype glioblastoma [44].

In conclusion, to date, the assessment of *TERT* in GBM *IDH*-wildtype has important diagnostic implications without a clear contribution to prognosis and treatment response prediction.

## 4. Development of TERT Inhibitors and Perspectives in GBM *IDH*-Wildtype Treatment

The development of TERT inhibitors is a very attractive possibility due to the high frequency of *TERTp* mutations, and their clonal distribution across cancer cells [11,45,46,47].

Nevertheless, the development of TERT inhibitors has long been hampered by a lack of molecular tridimensional and structural data.

Furthermore, there is a strong limitation related to the pre-clinical assessment of these compounds mainly due to the absence of in vivo models allowing an adequate evaluation [48].

To date, no therapies targeting TERT have been approved in clinical practice and few molecules have been tested [49]. This is mainly because we recently elucidated mechanisms associated with TERT activities in normal and neoplastic tissues. Furthermore, the toxicities of agents targeting TERT represent another issue limiting their investigation.

Moreover, despite TERT inhibition finally leading to cell cycle arrest and cell death in in vitro and in vivo models, this biological consequence occurs after reiterated cell divisions, necessary to achieve critically short telomere length [50].

Despite these potential limitations, several approaches have been explored in the development of targeted therapies in this field (Table 3).

The oligonucleotide inhibiting TERT enzymatic function, imetelstat (or GRN163L) binds the RNA template of human telomerase and acts as a competitive inhibitor of telomerase activity. Preclinical and clinical studies demonstrated a clinical efficacy of imetelstat in hematological malignancies including myelofibrosis, essential thrombocythemia, myelodysplastic syndromes, and acute myeloid leukemia.

In vitro, tumor-initiating cells isolated from primary GBM *IDH*-wildtype tumors and expanded as neurospheres showed a reduction in proliferation with imetelstat after approximately 15 to 20 population doublings. Again, the benefit of targeting TERT is delayed [56]. In vitro, promising results were obtained from the treatment of GBM *IDH*-wildtype neurosphere cells with imetelstat in association with ionizing radiation or temozolomide [56]

As regards off-target toxicity, normal brain tissue appears less susceptible to telomerase inhibition than brain cancer cells as the average telomere GBM *IDH*-wildtype length of GBM cells is shorter compared with normal human brain cells. Moreover, after the removal of imetelstat, telomerase activity is reversible to normal levels [56]. 

Unfortunately, on moving to the clinical phase, imetelstat resulted in very modest clinical activity in solid tumors, furthermore exposing patients to significant hematologic and hepatotoxic dose-limiting side effects [51].

In a phase 2 trial conducted in children with recurrent CNS tumors (*n* = 40), among grade 3/4 toxicities thrombocytopenia was registered in 32.5 % of patients, lymphopenia in 17.5 %, and neutropenia in 12.5 %. The study was closed after two patients died of intratumoral hemorrhage secondary to thrombocytopenia [57]. Currently, one phase 2 clinical trial is investigating imetelstat in younger patients with recurrent or refractory brain tumors (NCT01836549).

Given the previous disappointing results, further studies focused on alternative telomerase-dependent therapeutic approaches. To date, several strategies employing *TERTp* inhibition are under investigation [48].

Among telomerase inhibitors, BIBR1532 targets a critical site in the interaction between the hTR and TERT. However, its anti-tumor effects are burdened in vivo by water insolubility and low cellular uptake. Recent efforts to improve the release and efficacy of this small molecule led to promising results through Zeolitic imidazolate framework-8 (ZIF-8) as a delivery vehicle.

Inhibition of hTERT mRNA expression, cell cycle arrest, and increased cellular senescence were observed in cancer cells treated with the combination molecules [52]. However, they have not yet been tested on brain cancer cells.

A class of small molecules identified as potential telomerase inhibitors is constituted by G-quadruplex stabilizers. Folding of the 3′- overhang of telomeric DNA into G-quadruplex structures, which consist of a four-stranded helical guanine-rich DNA secondary structure, hampers access of telomerase to telomere ends. Molecules able to stabilize the telomeric G-quadruplex can cause telomere erosion and act as anticancer agents. Among these molecules, BRACO-19 induced viability loss in glioma cell lines, showing selectivity for cancer cells [53].

The 6-thio-2-deoxyguanosine (6-thio-dG) represents an interesting agent that has been employed as a drug able to restore or modify the immune-microenvironment. This compound can be incorporated into telomeres by telomerase in place of normal guanosine. This leads to telomere dysfunction due to interference with telomere structure and with telomere-binding proteins (e.g., shelterins). This mechanism, also known as the “telomere poisoning approach”, is linked to the onset of DNA-damage signals. Cells treated with 6-thio-dG release DNA fragments that are taken up by dendritic cells and activate a pathway involved in immune response, called STING (stimulator of interferon genes). In mouse models of colon cancer, lung cancer, and hepatocellular carcinoma (HCC), the 6-thio-dG induces immunogenic cell death [54]. This mechanism could be crucial to switching an immunologically cold tumor microenvironment into a hot tumor microenvironment [58,59]. Recent studies demonstrated the promising efficacy of a sequential administration of 6-thio-dG and anti-programmed death ligand 1 (PDL1) plus an inhibitor of the vascular endothelial growth factor (VEGF) in HCC [54]. To date, no studies are assessing the 6-thio-dG within GBM *IDH*-wildtype.

Another promising strategy under evaluation is the inhibition of transcription factors involved in TERT reactivation. In particular, the GABPβ1L subunit of the GABP transcription factor seems to be crucial to achieving the downregulation of telomerase in *TERTp* mutant cells. Disruption of GABPβ1L selectively inhibits TERT, and subsequent telomere loss favors cell death in *TERTp* mutant cancer cells. Moving from gene knockdown experiments, reduced GABPβ1L levels impaired tumor growth in vitro and extended survival in xenografted mice [60]. Similar preclinical models confirmed GABPβ1L as a potential therapeutic target through post-editing technologies, leading to cell death in vitro in 30 to 80 days [60,61]. In vivo, intracranial xenografts of GABPβ1L knockout cells exhibited reduced proliferation [61]. To overcome the time required to manifest the biological effect of telomerase inhibition, some authors hypothesized a synergistic effect between TERT inhibition and DNA-damaging agents, such as radiotherapy and cytotoxic chemotherapeutic agents. Amen A.M. et al. found knockout of GABP1L impaired the growth of *TERT* promoter mutant cells and reduced tumor growth rate in vivo, leaving normal cells unaffected. Furthermore, they demonstrated that loss of *TERT* activation sensitizes GBM *IDH*-wildtype to DNA damage. In particular, reduction in GABPB1L and administration of temozolomide had synergistic anti-tumor effects in vivo [62]. Inhibiting TERT makes cancer cells more sensitive to DNA breaks, due to the downregulation of DNA-damage repair mechanisms.

Aquilanti et al. provided further evidence about *TERT* reactivation and its role in tumor viability, beyond glioma initiation.

*TERT* promoter mutant-cells undergoing *TERT* knockdown exhibited features of telomere crisis and cell death, such as the formation of chromatin bridges and cell cycle arrest. Of note, cell death is achieved only after several cell divisions, necessary to cause telomere erosion and telomere crisis.

Considering the time required to see their biological effect, the optimal setting for a TERT inhibitor treatment could be after a gross total resection in an adjuvant setting more than in advanced disease in which a tumor response is generally required in a shorter time interval [50].

Some authors have suggested the possibility of targeting kinases upstream of the GABP-*TERT* axis. Since EGFR and TERTp are functionally connected, EGFR and AMPK could be tested as therapeutic targets in combination with other therapies to induce telomere reduction and tumor cell-killing [37]. No studies exploring this strategy are available. This also considers the negative results observed with agents targeting EGFR in GBM *IDH*-wildtype [63].

Over the past two decades, multiple studies have also evaluated telomerase-based immunotherapy and in particular vaccines, including peptide vaccines, dendritic cell vaccines, and DNA vaccines [64]. TERT is an intracellular protein that can be recognized by T cells after being presented on the external cell surface by major histocompatibility complex (MHC) molecules. In vitro, TERT protein has been demonstrated to be immunogenic for peripheral blood T lymphocytes: cancer cells present TERT peptides that can be recognized by either CD4^+^ or CD8^+^ T cells. In vivo, experiments demonstrated T-cell responses are in some cases associated with the inhibition of tumor growth.

Thus far, TERT-based vaccination has been studied in patients with different types of cancer in several phase 1 and 2 trials, and one phase 3 trial in pancreatic cancer patients. However, from a critical evaluation of these trials, it seems clear that therapeutic TERT-based vaccination induces temporary disease stabilization as the best response, with a poor effect on tumor size [65].

Regarding GBM *IDH*-wildtype, vaccination was tested in seven patients treated with a dendritic cell (DC)-based vaccine targeting GBM *IDH*-wildtype stem cells. An immune response was identified in all patients. No patients developed adverse autoimmune events or other side effects. Progression-free survival was 2.9 times longer in vaccinated patients compared to controls [66].

Recently, a DNA vaccine has been tested for safety and efficacy in a phase 1/2 trial in GBM *IDH*-wildtype patients. The trial enrolled 52 patients with newly diagnosed GBM *IDH*-wildtype, who were further divided into two cohorts (A: unmethylated MGMT and B: methylated MGMT). These patients received INO-5401 (synthetic DNA plasmid encoding hTERT, WT-1, PSMA) plus INO-9012 (synthetic DNA plasmid encoding IL-12), with cemiplimab (PD-1 inhibitor). Hypofractionated RT with temozolomide was administered to all patients, followed by maintenance therapy in Cohort B only. Most adverse events were ≤ grade 2, with no grade ≥ 4 events. Median OS in Cohorts A and B was 17.9 months and 32.5 months, respectively. The INO-5401 + INO-9012 has an acceptable risk/benefit profile and elicits robust immune responses that correlate with enhanced survival when administered with cemiplimab and RT/TMZ to newly diagnosed GBM *IDH*-wildtype patients [55].

Currently, a phase 2 study (NCT02818426) is evaluating UCPVax, a therapeutic anti-cancer vaccine based on the telomerase-derived helper peptides designed to induce strong TH1 CD4 T-cell responses in cancer patients. The study enrolls patients with GBM *IDH*-wildtype, pre-treated with standard radiochemotherapy.

## 5. Conclusions

*TERT* mutations assume a key role in driving development and progression in patients with GBM *IDH* wt. Due to its central biological role, a great deal of effort has been spent on the research of effective TERT inhibitors in these patients. Despite these aspects, the high-grade toxicities reported in clinical trials as well as the latency required by TERT inhibitors to achieve a biological effect are important limits to the development of effective drugs. However, an improved understanding of the TERT molecular structure and TERT interactions with other proteins have brought attention to the possible development of treatment strategies involving TERT selective inhibitors and other agents. In particular, the most promising early results come from combination therapies where TERT inhibition is combined with other approaches including immunotherapy. This last combination assumes a particular interest since inhibition of TERT could change the tumor-associated microenvironment toward an immune-active one. In conclusion, *TERT* inhibition is far from being a reality in clinical practice. Combination strategies employing these inhibitors are interesting opportunities despite still being in clinical trial investigation.

## Figures and Tables

**Figure 1 cells-13-00044-f001:**
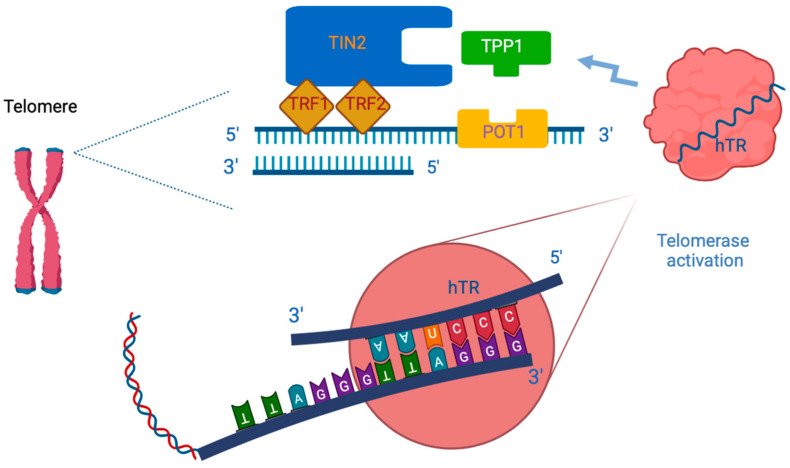
Telomerase recruitment through interaction with shelterin complex. TPP1—tripeptidyl peptidase 1; TRF1 = telomeric repeat-binding factor 1; TRF2 = telomeric repeat binding factor 2; TIN2 = TRF1-interacting nuclear factor 2; POT1 = protection of telomeres protein 1; hTR = RNA template of TERT [14].

**Figure 2 cells-13-00044-f002:**
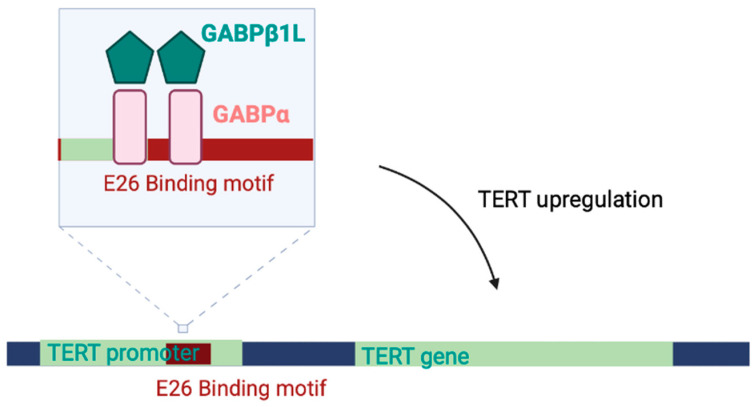
Regulation of *TERT* transcription through ETS transcription factor GABP. GABP tetramers, constituted by GABPα (DNA-binding subunit) and GABPβ1L (transactivating subunit) bind both the mutant E26 binding motifs and the normal motifs nearby. They are critical in activating the mutant *TERT* promoter.

**Table 3 cells-13-00044-t003:** Principal TERT inhibitors were discussed.

Agents under Investigation	Mechanisms of Action and Preliminary Results
Imetelstat [51]	Oligonucleotide acts as a competitive inhibitor of telomerase activity. No clinical benefit emerged in small clinical trials. Significant grade 3–4 thrombocytopenia. To date, a single trial investigating imetelstat in young patients with recurrent brain tumors (NCT01836549).
BIBR1532 [52]	A small molecule targeting the interaction between hTR and TERT. Water insolubility makes this drug difficult to manage. A specific delivery vehicle has been proposed (zeolitic imidazolate framework-8) to overcome this limit. To date, no evidence of efficacy within patients with GBM *IDH*-wildtype.
BRACO-19 [53]	Acts as G-quadruplex stabilizers modifying the DNA structure making difficult the binding of the telomerase. No studies on patients with GBM *IDH*-wildtype.
6-thio-2-deoxyguanosine [54]	An agent that can be incorporated into telomerase in place of normal guanosine leads to telomere dysfunction (“telomere poisoning approach”). This drug can mediate the modification of the tumor microenvironment from cold to hot in preclinical models. No studies on patients with GBM *IDH*-wildtype.
INO-5401 [55]	A synthetic DNA plasmid encoding hTERT. It has been administered with cemiplimab, and INO 9012 (synthetic DNA plasmid encoding IL-12). Administration of INO 5401 and INO 9012 resulted in an OS of 32.5 months in patients with methylated glioblastoma.

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
