# Peer review of "The Biological and Clinical Role of the Telomerase Reverse Transcriptase Gene in Glioblastoma: A Potential Therapeutic Target?"

_cells, 2023, doi:10.3390/cells13010044_

Round 1

Reviewer 1 Report

Comments and Suggestions for Authors

The authors present a review in which it is not clear whether the main review goal is to describe the TPM (tert promotor mutation) impact on survival or whether it is about noval drug discoveries. Given the abstract and titel, I suppose the main work needs to concentrate on novel compounds targeting TPM or TERT. However, D'Alessandris QG et al., Expert Reviews in Molecular Medicine 2023 made a much more comprehensive and very systematic review of noval drugs against TPM or TERT.  In this context, this review doesn't provide novel overviews. It does not even look for TPM inhibitors discovered in other cancer entities. This is why, this review should not be accepted.

Comments on the Quality of English Language

NA

Author Response

The authors present a review in which it is not clear whether the main review goal is to describe the TPM (tert promotor mutation) impact on survival or whether it is about noval drug discoveries. Given the abstract and titel, I suppose the main work needs to concentrate on novel compounds targeting TPM or TERT. However, D'Alessandris QG et al., Expert Reviews in Molecular Medicine 2023 made a much more comprehensive and very systematic review of noval drugs against TPM or TERT.  In this context, this review doesn't provide novel overviews. It does not even look for TPM inhibitors discovered in other cancer entities. This is why, this review should not be accepted.

We want to thank the reviewer for the comments provided. We specified better that this is a narrative review focused on TERT inhibitors in patients with Glioblastoma IDH-wild type. We also tried to discuss some issues related to the biological role of TERT on GBM IDH wt development and progression as well as the role of TERTp mutation on prognosis and clinical outcome.

In addition, we want to thank the reviewer for the suggested article carried out by D’Alessandris QG et al. We included this important systematic review in the text.

Reviewer 2 Report

Comments and Suggestions for Authors

The above review is very well written and executed. However, the authors have not cited a recently published review " Targeting Telomerase for cancer therapy by Adam N Guterres and Jessie Villanueva" which also focuses on the different Telomerase inhibitors and would complement this current draft. Another minor suggestion would be to elaborate on the existing shortcomings for testing these inhibitors, for example, the lack of in-vivo models for pre-clinical testing and differences between telomeres and telomerase between mouse and human system which further limits identification and devlopment of novel inhibitors.

Comments on the Quality of English Language

The review is very well written and can benefit from some proof-reading.

Author Response

The above review is very well written and executed. However, the authors have not cited a recently published review " Targeting Telomerase for cancer therapy by Adam N Guterres and Jessie Villanueva" which also focuses on the different Telomerase inhibitors and would complement this current draft. Another minor suggestion would be to elaborate on the existing shortcomings for testing these inhibitors, for example, the lack of in-vivo models for pre-clinical testing and differences between telomeres and telomerase between mouse and human system which further limits identification and devlopment of novel inhibitors.

We would thank the reviewer for the precious comment provided and the support shown for our work. We included and cited the suggested review of the text. We also specified better the limitations associated with the in-vivo assessment of these compounds.

Reviewer 3 Report

Comments and Suggestions for Authors

In this review, Vincenzo Di Nunno and co-workers provided an update on the biology of TERT.

The paper is sufficiently exhaustive and well-written. In my opinion, once it has been subjected to minor revisions, this work will be suitable for publication in “Cells”.

I suggest to review and expand the text of the Abstract and the Conclusions in order to make the aim of the paper more evident.

Minor comments:

·         Line 34: “telomers integrity” should be modified in “telomers’ integrity”.

·         Line 49: “TERTencodes” should be modified in “TERT encodes”.

·         Line 55: “GBM IDH wildty pe” should be modified in “GBM IDH wildtype”.

·         Line 55-59:  same concept reported twice with different bibliographical references.

·         Line 114: “TRF2 = =” remove “=”.

·         Line 119-120: several repetitions in the caption of Figure 1 should be removed

·         Line 207-211: the period should be reviewed and clarified

·         The formatting of table 2 should be standardized.

·         Line 258: “Table 2” should be modified in “Table 3”.

·         Line 301-302: the period should be reviewed and clarified.

Author Response

In this review, Vincenzo Di Nunno and co-workers provided an update on the biology of TERT.

The paper is sufficiently exhaustive and well-written. In my opinion, once it has been subjected to minor revisions, this work will be suitable for publication in “Cells”.

I suggest to review and expand the text of the Abstract and the Conclusions in order to make the aim of the paper more evident.

We would like to thank the reviewer for the interest shown in our work.  We improved the abstract and conclusion paragraphs in order to provide more data about the topic discussed in the present review and the conclusion reported.

Minor comments:

  • Line 34: “telomers integrity” should be modified in “telomers’ integrity”.

We corrected this typo.

  • Line 49: “TERTencodes” should be modified in “TERT encodes”.

Checked and corrected.

  • Line 55: “GBM IDH wildty pe” should be modified in “GBM IDH wildtype”.

We corrected this sentence.

  • Line 55-59: same concept reported twice with different bibliographical references.

We thank the reviewer for point this out. We removed this redundant sentence.

  • Line 114: “TRF2 = =” remove “=”.

We corrected this typo.

  • Line 119-120: several repetitions in the caption of Figure 1 should be removed

We removed redundant sentences.

  • Line 207-211: the period should be reviewed and clarified

We totally agree with this comment. We clarified this sentence.

  • The formatting of table 2 should be standardized.

We modified the format of table 2.

  • Line 258: “Table 2” should be modified in “Table 3”.

We modified the legend of the table as required.

  • Line 301-302: the period should be reviewed and clarified.

We thank the reviewer for this observation. We modified this sentence.

Round 2

Reviewer 1 Report

Comments and Suggestions for Authors

The fact of summarizing view noval agents is not enough for a comprehensive review about tert inhibitors in glioblastoma. Especially since other systematic reviews include those treatments. The outcome table is interesting but is not the main scope of the article. I therefore must conclude that the "Cells" Journal impact factor is to high for that review. 

Author Response

We want to thank the reviewer for the comment and the issue pointed out related to our paper. The main goal of our narrative review is to improve knowledge of the pathogenic role and clinical impact of TERT mutation in GBM patients. After this, we critically discuss limitations and possibilities related to TERT inhibition in this group of patients. Therefore, we decided to agree with the suggestion provided by the reviewer and we modified the title of our paper specifying that we want to investigate all these aspects more than the ‘’only’’ therapeutic issue.